# Chemotactic behaviour of *Escherichia coli* at high cell density

Remy Colin [1,2]*, Knut Drescher [1,2,3] & Victor Sourjik[1,2]*

At high cell density, swimming bacteria exhibit collective motility patterns, self-organized through physical interactions of a however still debated nature. Although high-density behaviours are frequent in natural situations, it remained unknown how collective motion affects chemotaxis, the main physiological function of motility, which enables bacteria to follow environmental gradients in their habitats. Here, we systematically investigate this question in the model organism *Escherichia coli*, varying cell density, cell length, and suspension confinement. The characteristics of the collective motion indicate that hydrodynamic interactions between swimmers made the primary contribution to its emergence. We observe that the chemotactic drift is moderately enhanced at intermediate cell densities, peaks, and is then strongly suppressed at higher densities. Numerical simulations reveal that this suppression occurs because the collective motion disturbs the choreography necessary for chemotactic sensing. We suggest that this physical hindrance imposes a fundamental constraint on high-density behaviours of motile bacteria, including swarming and the formation of multicellular aggregates and biofilms.

---

[1] Max Planck Institute for Terrestrial Microbiology, Karl-von-Frisch-Strasse 10, Marburg, Germany. [2] Loewe Center for Synthetic Microbiology, Karl-von-Frisch-Strasse 16, Marburg, Germany. [3] Fachbereich Physik, Philipps-Universität Marburg, Karl-von-Frisch-Str. 16, 35043 Marburg, Germany. *email: remy.colin@synmikro.mpi-marburg.mpg.de; victor.sourjik@synmikro.mpi-marburg.mpg.de

When the cell density of a suspension of swimming bacteria increases, collective motion emerges, characterized by intermittent jets and swirls of groups of cells[1–3]. This behaviour is observed for many microorganisms not only in artificial but also in natural situations, often at an interface, e.g., when bacteria swarm on a moist surface in the lab[4–8] or during infection[9], or at an air-water interface during formation of pellicle biofilms[1,10]. Bacterial collective motion has been extensively studied experimentally[11–14] and theoretically[15–20], and it is known to emerge from the alignment between the self-propelled cells[21]. Two alignment mechanisms have been proposed, based either on steric interactions between the rod-like bacteria[22–24] or on the hydrodynamics of the flow they create as they swim[15,17], which displays a pusher force dipole flow symmetry[3,25,26]. However, the relative importance of these two mechanisms has not been clearly established so far[27].

Bacterial collective motion contrasts to the behaviour of individual motile cells in dilute suspension, when bacteria swim in relatively straight second-long runs interrupted by short reorientations (tumbles), resulting at long times in a random walk by which they explore their environment[28]. Bacteria can furthermore navigate in environmental gradients by biasing this motion pattern: they lengthen (resp. shorten) their runs when swimming toward attractive (resp. repulsive) environment[28]. The biochemical signalling pathway controlling this chemotactic behaviour is well understood in E. coli[29,30] and it is one of the best modelled biological signalling systems[31]. Bacteria monitor—via their chemoreceptors—the changes in environmental conditions and respond to them by modifying a phosphorylation signal transmitted to the flagellar motors to change the tumbling frequency accordingly[32,33]. In E. coli, attractant substances repress the phosphorylation signal, which results in prolonged runs, repellents having the opposite effect. An adaptation module slowly resets the receptor sensitivity for further stimulations, via a negative feedback loop[34,35]. This effectively allows the cell to compare its current situation to the recent past while swimming along a given direction[30], with a memory time scale of a few seconds[36,37]. Notably, the rotational Brownian motion of the cell body interferes with this mechanism of sensing by randomly changing the direction of swimming while temporal comparisons are performed[28,29].

Although typically seen as a single-cell behaviour, chemotaxis also drives collective behaviours based on chemical interactions, such as autoaggregation[38,39], self-concentration in patches[40,41] and travelling band formation[42,43], where the chemotactic response to self-generated gradients of chemoattractants leads to local cell density increases. However, very little is known about how the high-density physical interactions and the resulting collective motion influence the chemotactic navigation of bacteria[44,45]: for example, it is unclear whether chemotaxis would be improved by alignments of convective flows with the gradient[19,45] or instead compromised by random collisions between cells. This lack of knowledge is in part due to the technical difficulty of measuring the dynamics of cells in a dense suspension[20]. Over the last few years, new image analysis methods have been developed or adapted to bacterial systems[2,46–49], which exploit intensity fluctuations[46] or intensity pattern shifts[49] to characterize swimming and chemotaxis in populations of bacteria. These Fourier image analysis methods function at arbitrarily high cell densities, capture the dynamics of all cells without bias, and are at least as fast as the more commonly used particle tracking techniques.

In this paper, we use Fourier image analysis methods to investigate how the collective motion developing with increasing cell density affects the ability of E. coli populations to follow controlled chemical gradients in a microdevice. Our experimental results and computer simulations show that, after increasing up to a maximum at intermediate densities, chemotaxis is strongly reduced as collective motion developed in the sample. Collective reorientations act similarly to an active rotational diffusion interfering with the chemosensing mechanism. Additionally, the characteristics of the collective motion are consistent with hydrodynamic interactions being the primary driver of its emergence, additional steric effects being important but secondary. These results have important implications for collective behaviours of motile bacteria at high density.

## Results

**Measuring chemotactic motion at variable cell density**. To measure both the collective dynamics and the chemotactic response of populations of E. coli cells at varying density, we analyzed bacterial swimming in controlled gradients of a chemical attractant using a microfluidics setup, videomicroscopy and Fourier image analysis. Microfabricated devices made of the oxygen-permeable polymer poly-dimethylsiloxane (PDMS) were used as described previously[49] to create quasi-static linear gradients of the non-metabolizable attractant α-methyl-D,L-aspartic acid (MeAsp) in a straight channel (Fig. 1a). This gradient could be mimicked using the fluorescent dye fluorescein (Fig. 1b), which has a diffusion coefficient similar to the one of MeAsp ($D \simeq 500 \ \mu m^2/s$). Three channel heights, $h = 50$, 30 and 8 $\mu m$, were used to increasingly confine the cell suspensions towards two dimensions. Motility of suspensions of E. coli bacteria, measured in the middle of this gradient, was sustained for several hours at all cell densities (Supplementary Fig. 1a), due to oxygen availability and abounding energy source in the medium. When the density increased, the development of collective motion in the sample, characterized by the existence of collective eddies and jets of swimming cells was observed (Fig. 2a) as expected[2,13,20]. The shape of the gradient, after the transient phase during which it becomes established, is largely unaffected by the collective motion (Fig. 1c), except at the highest bacterial cell body volume fractions. Consistantly, the Peclet number, which compares diffusion and advective transport of MeAsp in an eddy, remains moderate ($Pe = hv_f/D \leq 1$, where $v_f$ is the fluid velocity; see Supplementary Note 1), confirming that stirring by the bacteria has little effect on the diffusion of MeAsp[50]. Furthermore, in the present geometry, at steady state the gradient shape does not depend on the diffusion coefficient. Because the cell density slowly changes with time in the area of measurement due to the chemotactic accumulation, thus differing from the average cell density in the suspension, the cell body volume fraction $\Phi_c$ was measured in situ for each experiment (Supplementary Fig. 2 and "Methods").

To further investigate the effect of cell elongation, which is typically observed for bacteria swarming on a wet hydrogel at high density[4], we compared normally-sized E. coli with average length $L = 2 \ \mu m$ and cells elongated to an average of $L = 4 \ \mu m$ upon treatment with cephalexin, a drug preventing cell division ("Methods" and Supplementary Fig. 2). We observed that the cephalexin treatment strongly increases cell autoaggregation, which is mediated in this E. coli strain by the adhesin Antigen 43[38]. We then used Δflu strain deleted for the gene encoding this adhesin for all results reported in this study, since, in absence of the cephalexin treatment, flu+ and Δflu cells behave similarly, regarding both their collective motion and chemotaxis (Supplementary Fig. 3), and the elongated Δflu cells show normal motility (Supplementary Fig. 1).

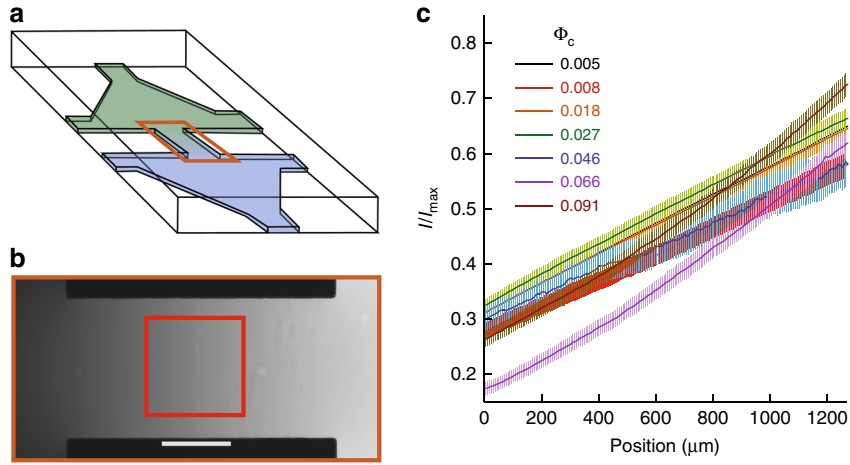

**Fig. 1** Microfluidic device used for chemotaxis assays. **a** Schematic representation of the device, where two reservoirs with different chemical composition are connected by a channel in which the chemoattractant gradient forms. **b** Central part of the device, highlighted orange in (**a**), with gradient profile across the channel quantified using fluorescein. Scale bar is 500 $\mu$m. The red box indicates the location at which cellular behaviour is recorded. **c** Examples of gradients, measured in the $h = 50$ $\mu$m device, for cell suspensions of indicated cell densities. Error bars represent measurement error.

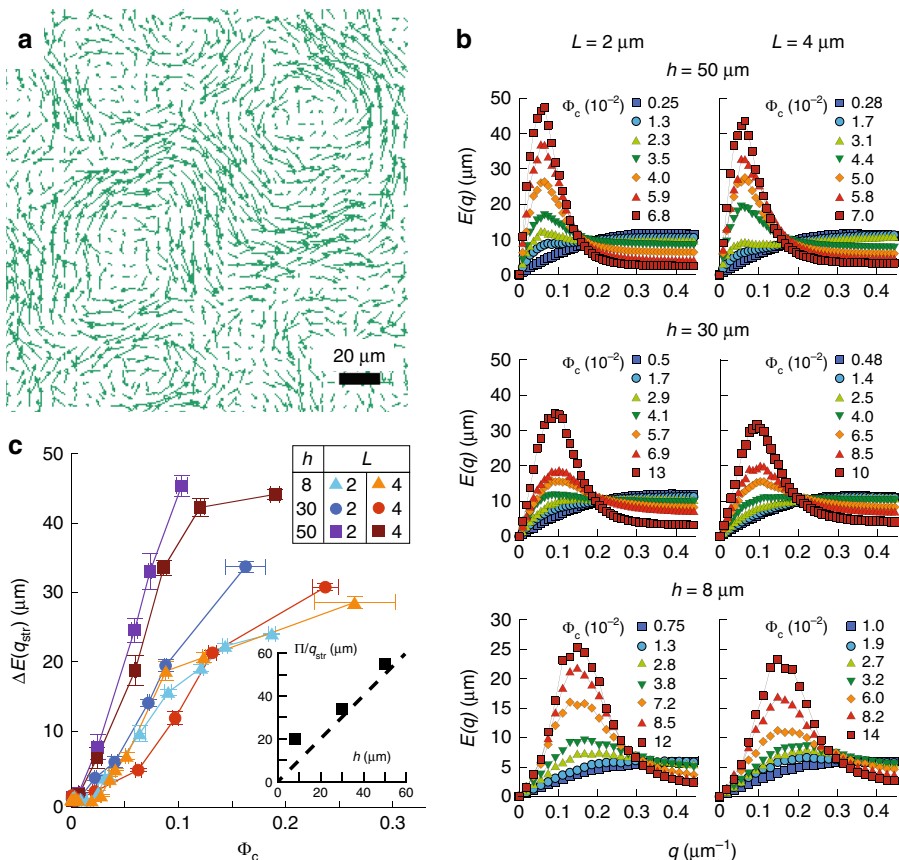

**Fig. 2** Collective motion in bacterial suspensions. **a** Typical snapshot of the velocity field measured in a high-density cell suspension ($\Phi_c = 0.08$, cell length $L = 2$ $\mu$m, channel height $h = 50$ $\mu$m). **b** Flow structure factor $E(q)$ for increasing cell densities at the indicated values of channel height and cell length. **c** Amplitude of the peak, corrected for the low-density value, $\Delta E(q_{str})$ as a function of cell density in the different experimental conditions. Each point is the median, and associated error bar the standard error of the mean (SEM) of 8 measurements, binned according to cell density. Inset: $\pi/q_{str}$ as a function of channel height. Dashed line indicates equality.

**Structure of the collective motion**. The collective motion was measured in the gradient using an image velocimetry method derived from Phase Differential Microscopy[49] ("Methods") to quantify the local velocity field $\mathbf{v}(\mathbf{r})$, averaged over several neighbouring bacteria (Fig. 2a). The spatial structure of the flow field at varying cell volume fraction $\Phi_c$ was characterized by the

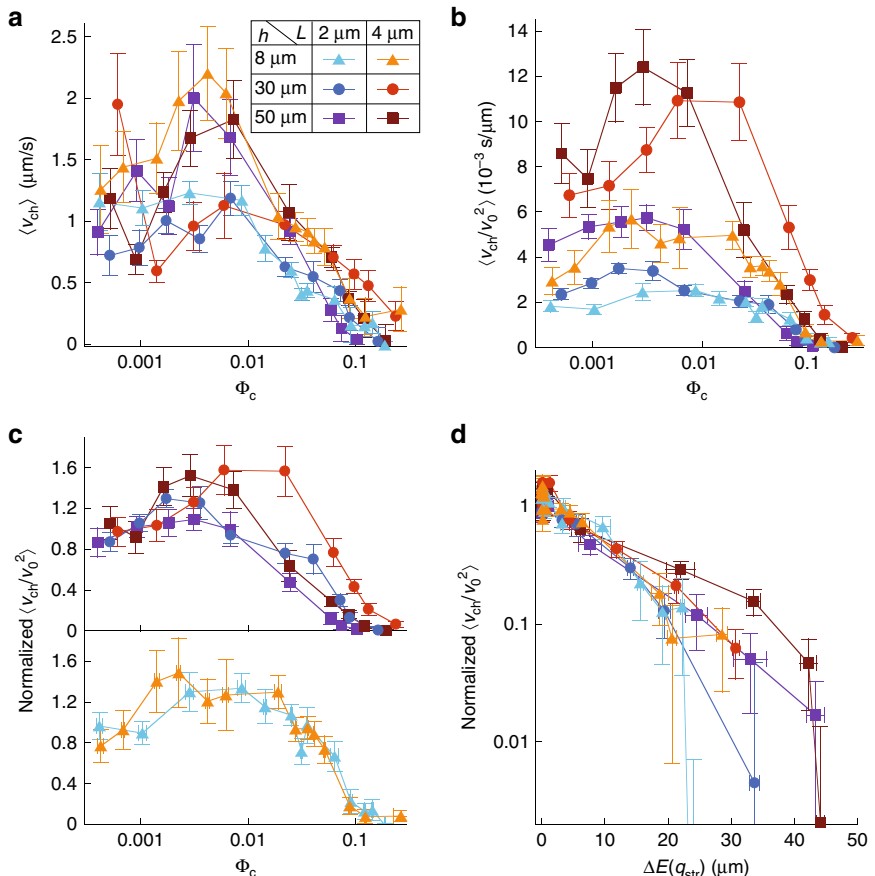

**Fig. 3** Effect of collective motion on the chemotactic drift. **a**, Chemotactic drift $v_{ch}$ as a function of cell density for indicated values of cell length (L) and channel height (h). **b** Chemotactic coefficient $v_{ch}/v_0^2$ as a function of cell density. **c–d** Chemotactic coefficient normalized to its low-density value as a function of cell density (**c**) or of the amplitude $\Delta E(q_{str})$ of the collective motion (**d**). Labels throughout the figure are defined in panel (**a**). Each point represents the median of 8 experiments. Error bars represent the SEM in abscissa and the sum of the SEM and the mean measurement error in ordinate.

2-dimensional power spectral density of the velocity field ("Methods")[20]:

$$E(q) = q \left\langle \frac{\tilde{\mathbf{v}}(\mathbf{q},t)\tilde{\mathbf{v}}^*(\mathbf{q},t)}{A_0 \langle \mathbf{v}^2 \rangle} \right\rangle_{t,|\mathbf{q}|=q}, \quad (1)$$

with $\tilde{\mathbf{v}}$ the spatial Fourier transform of $\mathbf{v}$, $\mathbf{q}$ the wave vector and $A_0$ the area of the field of view. This quantity $E(q)$ (Fig. 2b) measures the distribution of kinetic energy over the flow structure sizes $\pi/q$, hence representing the probability to observe a "vortex" of size $\pi/q$[20]. At $\Phi_c \gtrsim 0.01$, this flow structure factor exhibits, at a low wave number $q_{max}$, a maximum $E(q_{max})$, which grows in amplitude as the cell volume fraction increases (Fig. 2b). After an initial decrease, the value of $q_{max}$ reaches a plateau $q_{str}$ at moderate cell volume fraction, corresponding to $E(q_{max}) \gtrsim 15~\mu m$ (Supplementary Fig. 4a). Fully developed collective motion thus takes the form of flow structures (eddies) with a specific size, $\pi/q_{str}$. An increase in cell density results in an increase of the amount of collective flow of this size – the amplitude of the collective motion, which can be quantified by $E(q_{str})$. Interestingly, $q_{str}$ is apparently determined by the height $h$ of the channel (Fig. 2c inset and Supplementary Fig. 4a), indicating that the flow structures are constrained by the system size. The observed decrease of $q_{max}$ at low cell volume fraction might be due to a combination of $E(q_{max})$ reaching the noise level, represented by the featureless $E(q)$ observed at the lowest cell volume fractions

(Fig. 2b), and a genuine reduction of the eddy size when $\Phi_c$ becomes low.

For simplicity, the amplitude of the collective motion was quantified by $\Delta E(q_{str})$ at all volume fractions (Fig. 2c), i.e., $E(q_{str})$ (Supplementary Fig. 4b, c) corrected for background noise. We observed that $\Delta E(q_{str})$ tends to grow more slowly with volume fraction for more confined suspensions (lower $h$). At moderate confinement ($h = 30$ and $50~\mu m$), $\Delta E(q_{str})$ also grows more slowly for longer cells, whereas for $h = 8~\mu m$ it is on the contrary a single function of $\Phi_c$ for both cell lengths (Fig. 2c). Importantly, when normalized to its value at high $\Phi_c$, $\Delta E(q_{str})$ was found to be for all conditions a single function of the flux of bacterial mass integrated over the vortex size $\Phi_c v_0 \pi/q_{str}$, where $v_0$ is the population-averaged swimming speed of the cells (Supplementary Fig. 4d). For $\Phi_c \lesssim 0.01$, $v_0$ itself is on average constant at a value between 10 and 25 $\mu$m/s which depends on cell length and the degree of confinement, and also tends to vary between biological replicates (Supplementary Fig. 1b, c). At $\Phi_c \gtrsim 0.01$, $v_0$ progressively increases, by a factor of up to two, as $\Delta E(q_{str})$ grows and the collective motion developed, consistently with previously reported behaviour[2].

**Dependence of chemotaxis on cell density.** The chemotactic response of the cells to the MeAsp gradient was first quantified using their chemotactic drift velocity $v_{ch}$, i.e., the population-averaged speed of displacement up the gradient, measured as

detailed in the "Methods" section following Colin et al.[49]. The gradient of MeAsp concentration (from 0 to 200 μM in 2 mm) was chosen to maximize $v_{ch}$ at low cell density[49]. For all combinations of cell length $L$ and channel height $h$, we observed that the chemotactic drift $v_{ch}$ first tends to increase when the cell volume fraction $\Phi_c$ increases in the range $5 \cdot 10^{-4} - 0.01$. It then strongly decreases above a cell volume fraction $\Phi_c$ that depends on $L$ and $h$, corresponding to the one above which collective behaviour is observed (Fig. 3a). Consistently, the drift decreases as function of $\Delta E(q_{str})$, albeit differently for each $L$ and $h$ (Supplementary Fig. 5). Swirling collective motion thus clearly impairs the ability of the cells to perform chemotaxis.

Since the chemotactic drift depends both on the cell swimming speed and on their ability to bias their direction of motion towards the gradient—the chemotactic efficiency, we aimed to separate these two contributions by normalizing $v_{ch}$ to the swimming speed $v_0$. For non-interacting cells in 3D in a chemical gradient, they are expected to be related by[51–53]:

$$v_{ch} = G v_0^2 \tau(\tau_m, \tau_T, \tau_R) \nabla f(c) , \quad (2)$$

where $G$ is the total gain of the chemotaxis system, $f(c)$ is the part of the chemoreceptor free energy difference due to the binding of chemical ligand, present at concentration $c$ and $\tau$ is the typical time during which the bacterium is able to measure a change of $f(c)$ in a given direction. The latter depends on the memory time scale $\tau_m$ as well as on two reorientation times. The first is due to Brownian rotational diffusion during the runs, $\tau_R = 1/D_r$, with $D_r$ the rotational diffusion constant of isolated cells. The other is due to tumbling, $\tau_T = \tau_0/(1 - \exp(-D_T\tau_t))$, with $\tau_0$ the steady state tumbling rate and $D_T\tau_t$ the mean squared angular change during tumbles. It is expected from previous studies[51,52,54] that (Supplementary Note 2):

$$\tau = \frac{\tau_R}{\tau_R + \tau_T} \frac{\tau_T}{1 + \tau_T/\tau_R + \tau_T/\tau_m} . \quad (3)$$

In absence of interactions, according to Eq. 2, the chemotactic coefficient $v_{ch}/v_0^2$ depends only on internal—fixed—parameters of the chemotaxis pathway and on the fixed concentration profile. This quantity was therefore chosen to quantify chemotactic efficiency at all densities, although at high densities cell behaviour might deviate from Eq. 2. We observed that the chemotactic coefficient $v_{ch}/v_0^2$, like the chemotactic drift, tends to increase with volume fraction, peaks at a condition–dependent $\Phi_c$ around 0.01, before decreasing sharply —by a factor of up to 100—as the collective motion develops (Fig. 3b). We therefore concluded that both the intermediate-maximum of $v_{ch}$ and its subsequent reduction by the collective motion arise from cell-density effect on the chemotactic efficiency, and that the moderate increase in swimming speed at high cell density cannot compensate for the much stronger density dependence of the chemotactic coefficient.

Chemotaxis was also found to be affected by cell length and channel height. Even at low cell volume fraction, if cells are longer or in a higher channel, they have a higher chemotactic coefficient $v_{ch}/v_0^2$ (Fig. 3b). Cell elongation is indeed expected to result in increased chemotactic coefficient in steady gradients[55], because of the lower Brownian rotational diffusion of longer cells[51] and their expected reduced tumbling angle[42,56]. To directly compare the pure effect of cell density under various $L$ and $h$, chemotactic coefficients were normalized to the average value at low volume fractions ($\Phi_c < 0.0015$) in each condition (Fig. 3c). Even this normalized chemotactic coefficient is significantly higher for longer cells at $\Phi_c > 0.01$ under moderate confinement ($h = 50$

and 30 μm, Fig. 3b). It is however a single function of $\Phi_c$, irrespective of cell length, for $h = 8$ μm (Fig. 3c).

The behaviour of the chemotactic coefficient is thus reminiscent of the one of $E(q_{str})$. Indeed, the normalized chemotactic coefficient was found to be a single function of the amplitude of the collective motion $\Delta E(q_{str})$, irrespective of the cell length and channel height (Fig. 3d), decreasing exponentially as a function of $\Delta E(q_{str})$ from its low-density value. The amplitude of the collective motion therefore appears to determine the reduction of the ability to follow gradients, irrespective of vortex size $\pi/q_{str}$, for $\Phi_c \gtrsim 0.01$, in all conditions.

**Collective reorientations impair chemotaxis.** As the indirect effect of gradient distortion by the swimming bacteria could be ruled out (Fig. 1), we hypothesized that the observed reduction of chemotaxis at high density is due to the forced reorientations resulting from the physical interactions, which would interfere with the sensing mechanism based on temporal comparisons: As the cell is swimming, it monitors the change in chemoattractant concentration within a few seconds, to decide whether to tumble. If during this time the direction in which the cell swims has changed significantly, the decision becomes less relevant, thus making the biochemically hard-wired bacterial chemotaxis strategy inefficient. In this hypothesis, collective reorientations would then act analogous to an enhancement of rotational diffusion, where the time scale of sensing $\tau$ would decrease because $D_r$ effectively increases[52,53,57], reducing the chemotactic coefficient accordingly (Eqs. 2 and 3).

To test whether collective reorientations may indeed reduce chemotactic drift, we performed numerical simulations of a population of self-propelled chemotactic rods of variable aspect ratio $L$ in a viscous medium, with the cell motion being confined to two dimensions. The cells interact sterically upon contact as well as hydrodynamically (Fig. 4a), being transported and reoriented by the flow generated by the other cells, which were approximated as pusher force dipoles in Hele-Shaw geometry[8,58]. The strength and reach of the hydrodynamic interaction is channel-height dependent in this approximation because of viscous friction on the walls (Methods and Supplementary Fig. 6). Rods reorient rotationally when tumbling, with the tumbling probability being determined according to the concentration of chemoattractant experienced by the rod, using a model of the chemotaxis system of *E. coli*[59]. They also are subject to rotational Brownian motion, with the rotational diffusion coefficient $D_r$ being smaller for longer cells[55].

When the cell area fraction increases in the simulations, the rods form increasingly large swirling packs of cells, similarly to previous reports, whether hydrodynamic interactions are present[8,14,20,58] (full simulations, Fig. 4b) or not[23,24] (steric-only simulations, Supplementary Fig. 7). However, qualitatively better agreement with experiments is achieved when hydrodynamic interactions are taken into account, suggesting that they are the primary contributor to the emergence of the collective motion in this case. Notably, when hydrodynamics is accounted for, the flow structure factor $E(q)$ peaks at a $q_{str}$ which depends only on channel height and not on cell density or cell length, in agreement with experiments (Fig. 4c), although the simulated vortex size is smaller than the experimental one. In contrast, when only steric interactions are simulated, $q_{str}$ clearly depends on cell length (Supplementary Fig. 7d). The dependences of simulated $E(q_{str})$ on channel height, cell length and cell density in the full simulations are similar to the experiments (Fig. 4d). Similarly, the swimming speed $v_0$ increases in the full simulations, although to a lesser extent than in the experiments, whereas it decreases in the

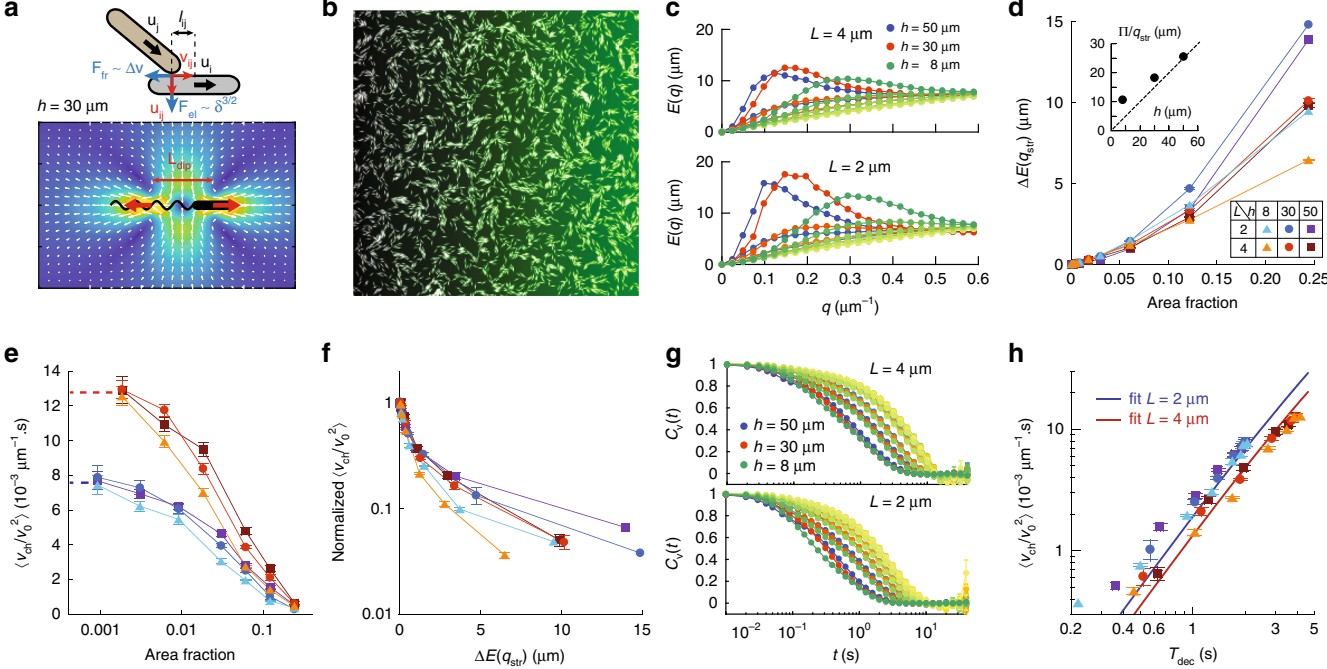

**Fig. 4** Numerical simulations of self-propelled interacting chemotactic rods in 2D. **a** Schematic representation of the simulated steric interaction between the rods, where interpenetration ($\delta$) results in Hertzian-like repulsion ($F_{el}$) and friction ($F_{fr}$), and the fluid flow generated by the hydrodynamic force dipole, here for channel height $h = 30\ \mu m$. For other heights, see Supplementary Fig. 6. **b** Snapshot of a simulation output (length $L = 4\ \mu m$, $h = 30\ \mu m$, $\Phi = 0.244$), showing collective motion of packs of rods. The green shading represents the gradient. **c** Flow structure factor $E(q)$ for indicated conditions, exhibiting a growing maximum at a $q_{str}$ depending only on the channel height $h$. Darkening shading of a given colour indicates increasing area fraction in the range $0.002 - 0.25$. **d** Maximum subtracted of its low-density value, $\Delta E(q_{str})$, as a function of cell area fraction. (Inset) Typical vortex size $\pi/q_{str}$ as a function of the channel height. Note the difference in scale compared to Fig. 2c. **e** Chemotactic coefficient as a function of the cell area fraction. Dotted lines represent its value in absence of interactions. **f**, Chemotactic coefficient, normalized to its low-density value, as a function of the maximum of the flow structure factor $\Delta E(q_{str})$. **g** Time autocorrelations of the cells velocity, the colour coding is the same as in (**c**). **h** Chemotactic coefficient as a function of the decorrelation time $\tau_{dec}$, defined by $C_v(\tau_{dec}) = 0.5$, and fit according to Eq. (4). **d**–**f**, **h**, Conditions are as indicated in panel **d** and error bars represent SEM on at least 3 runs, totalizing at least 1000 cells.

steric-only case (Supplementary Fig. 8), suggesting that this increase is due to hydrodynamic entrainment and further confirming the importance of hydrodynamic interactions in the emergence of the collective motion.

In the full simulations, the chemotactic coefficient $v_{ch}/v_0^2$, plotted as a function of cell body area fraction, decreases at the cell densities where the collective motion develops (Fig. 4e). However, contrary to the experiments, no increase in chemotactic coefficient is observed at intermediate cell area fractions. Nevertheless, $v_{ch}/v_0^2$ does scale with $E(q_{str})$, as in the experiments, although with a sharper decrease (Fig. 4f). In the steric-only simulations, $v_{ch}/v_0^2$ instead peaks as a function of cell area fraction at high densities but it is not a single function of $E(q_{str})$ (Supplementary Fig. 7e).

Finally, we took advantage of having access to all single-cell trajectories in the simulations to gain more insight into the mechanism of chemotactic drift reduction. The time autocorrelation of the individual cell velocity $\mathbf{v}_i$, $C_v(t) = \langle \mathbf{v}_i(t + t_0) \cdot \mathbf{v}_i(t_0) \rangle_{i,t_0} / \langle \mathbf{v}_i^2(t_0) \rangle_{i,t_0}$, shows a faster decorrelation as cell density increases for all values of $h$ and $L$, because of the collective reorientations (Fig. 4g). We used the decorrelation time, defined as $C_v(\tau_{dec}) = 0.5$, as a measure of directional persistence (Supplementary Fig. 9a). In absence of all physical interactions, the decorrelation time depends on the Brownian and tumbling reorientation times as $1/\tau_{dec} = 1/\tau_R + 1/\tau_T$, and combining Eqs. 2 and 3, the chemotactic coefficient can be

written as:

$$\langle v_{ch}/v_0^2 \rangle = \frac{\tau_{dec}^2/\tau_T}{1 + \tau_{dec}/\tau_m} G \nabla f(c) . \tag{4}$$

At higher cell densities, $\tau_{dec}$ also includes collective reorientations, $1/\tau_{dec} = 1/\tau_R^{eff} + 1/\tau_T$, with $1/\tau_R^{eff} = 1/\tau_R + 1/\tau_C$ an effective rotational diffusion rate including Brownian ($\tau_R$) and collective ($\tau_C$) reorientation times. Despite this additional contribution of collective motion, the dependence of the simulated chemotactic coefficient as a function of $\tau_{dec}$ is found to follow Eq. 4 very well at all densities, with the computed—and not fitted—coefficient values given in Supplementary Table 2, for all conditions in the full simulations (Fig. 4h). The effect of simulated collective motion can therefore be interpreted as an active enhancement of rotational diffusion, which decreases $\tau_R^{eff}$, hence $\tau_{dec}$ and the drift. In contrast, the chemotactic coefficient does not match well the dependence on the reorientation time predicted by Eq. 4 when only steric interactions are taken into account (Supplementary Note 3 and Supplementary Fig. 6h). Notably, the reorientations coming from physical cell-cell interactions do not necessarily respect the detailed balance of cell orientation fluxes, as Brownian rotational diffusion does, which may explain the deviations from Eq. 4 that we observed even when the hydrodynamic interactions are taken into account (Supplementary Note 4 and Supplementary Figs. 9 and 10).

## Discussion

Although the principles of bacterial chemotactic sensing are fairly well understood for a single cell[29,31], little was known about the effects of physical interactions between cells on chemotaxis, despite their frequent occurrence during self-concentration processes[38–43] and high-density collective motility[4,5,10]. While the physical properties of the collective motion, e.g. in a swarming colony, are largely unaffected by activity of the chemotaxis pathway[58,60,61], the reverse is not necessarily true. Here, we thus used suspensions of *E. coli* in a controlled environment as a model system to investigate the effect of collective motion, emerging when cell density increases, on chemotactic sensing.

We observed that the size of bacterial flow structures in fully developed collective motion is set by the smallest system size, the channel height, independently of cell length and volume fraction, and that an increasing amount of the kinetic energy of the system gets poured into this flow structure as the cell density increases. This property strongly suggests that hydrodynamics plays the primary role in the emergence of the collective motion, since, in the Hele-Shaw geometry of our system, the channel height sets the reach of the hydrodynamic interaction[8], fundamentally because of viscous friction on its top and bottom walls. Previous works also predicted that the largest possible flow structure—set by system size—dominates in hydrodynamics-based collective motion[15,62]. Also consistent with previous simulations of such motion[63] is the observed reduction of the total kinetic energy at fixed $\Phi_c$ but under increased confinement. Interestingly, similar flow properties are observed for the hydrodynamics-based collective flows during sedimentation of dense suspensions of passive particles[64,65]. Consistantly, our numerical simulations show that considering hydrodynamic interactions is key to reproduce the main experimental features, including the dependence of the vortex size on channel height, and its independence of cell length. Although the simulated vortex size is smaller than in the experiments, such shifts were previously observed on other quantities for this type of model[58]. Notably, besides the channel height, hydrodynamic interactions also set another characteristic length in our system, the hydrodynamic dipole length. This might explain the apparent saturation of the vortex size at about 20 μm in the experiments, when the channel height approaches the estimated dipole length (∼6 μm).

Physical interactions between cells at high densities result in a strong reduction and ultimately in the abolishment of the specific ability of *E. coli* to track chemical gradients, and thus of the chemotactic drift, despite the moderate increase in swimming speed due to collective entrainment. The collective motion is the driver of this decrease, with its amplitude $E(q_{str})$ being the sole determinant of reduced chemotactic efficiency. Collective motion acts directly on the mechanism of gradient-sensing since the gradient itself is little affected. Our analysis based on agent-based simulations suggests that this reduction is induced by an increased reorientation rate of cell bodies due to the collective motion, whether it emerges from steric or hydrodynamic interactions, which in this sense acted similarly to an active rotational diffusion. Importantly, we conclude that Eq. 2 derived for non-interacting swimmers can describe the chemotactic drift well at all densities, provided that the time scale of rotational diffusion is set to account not only for Brownian but also for interaction-induced reorientations. A similar reduction of chemotaxis by forced cell reorientations could be expected in other contexts, such as cells swimming in circles near surfaces[66] or during migration through a porous medium where cells collide with obstacles[67,68].

In contrast to the inhibition at high densities, the chemotactic drift is enhanced between low and intermediate densities. Although the nature of this increase remains to be elucidated, it is unlikely to result from self-attraction[38,39] or other chemical effects, as its extent depends on the degree of confinement and on cell length. In the simulations, a similar enhancement is observed when only steric interactions are considered—but not with the full model including hydrodynamic interactions, although it is not clear whether the nature of the transient enhancement in those simulations is the same as in the experiments. This discrepancy, along with the other quantitative differences between experiments and simulations, could potentially be explained by a number of factors our hydrodynamic simulations do not account for, such as collisions with the top and bottom channel walls and other physical effects neglected by two-dimensional confinement, flagellar entanglements and fluid flows affecting the flagellar bundles stability[69], as well as the point force approximation.

The observed regulation of chemotactic behaviour through physical interactions among motile cells has several important consequences for bacterial high-density behaviours. First, it provides a physical mechanism that might regulate chemotactic accumulation of bacteria near sources of chemoattractants (e.g., nutrients), because gradually increasing cell density[70] would initially promote and subsequently limit the process. Indeed, this effect could explain why the density of cells entering a capillary filled with chemoattractant saturates as a function of the cell density in the suspension[71]. The density for which the chemotactic drift is maximal, $\Phi_c \simeq 0.01$, which should play a cut-off role, is indeed the typical maximal cell density reached within travelling chemotactic bands which form through a self-generated gradient[42,43]. Thus, the hitherto neglected effects of physical interactions should be taken into account when describing these phenomena, in conditions for which the density gets high. Second, the observed strong reduction in chemotactic drift at cell densities typical of swarming ($\Phi_c \sim 0.30$)[5] suggests that, without specific counteracting mechanisms, chemotactic navigation of bacteria swimming within a swarm is nearly impossible, consistent with recent indications that the swarm expansion rate is set by the cell growth rate rather than motility[8]. Interestingly, we observed that cell elongation, one of the major hallmarks of swarming bacteria[4], indeed improved chemotaxis at high $\Phi_c$ under moderate confinement. However, it appeared to have little effect under stronger confinement expected in the swarm. Bacterial swarming was already known to be unaffected by the lack of functional chemotactic sensing[60,61]. Although more prominent steric interactions within a swarming colony[7,8] might potentially improve tracking of gradients at high density, as could other differences in swimming behaviour[72,73], or additional cohesive interactions[44] caused by cell differentiation in a swarm, our results suggest that the emergence of swirling collective motion fundamentally undermines the chemotactic behaviour.

## Methods

**Strains and cell culture**. *Escherichia coli* strain W3110 (RpoS+), *flu+* or Δ*flu*[38], were grown at 34 °C in Tryptone Broth (TB) from a 100-fold dilution of overnight culture to an optical density at 600 nm $OD_{600} = 0.7$. Where applicable, the culture was supplemented with 0.01 % cephalexin at one division time (1 h) before harvesting the cells. Cells were washed thrice and resuspended in motility buffer (10 mM $KPO_4$, 0.1 mM EDTA, 67 mM NaCl, pH 7.0) supplemented with 55 mM glucose as energy source, cooled to 4 °C for 20 min to reduce metabolic activity and then concentrated by centrifugation (1.5 $10^3$ g, 7 min) to a final cell density of $\Phi_c \sim 0.20$ ($OD_{600} \sim 100$). Series of dilutions were then performed in motility buffer, so that the amount of glucose per cell remains constant for all cell densities (∼0.5 mM/$OD_{600}$).

**Microfabrication**. Molds were fabricated using standard photolithography and microfabrication techniques. The SU8 photoresist (Microchem™) was spincoated on a silicon wafer for 90 s, covered with a positive mask of the device, produced using AutoCAD and printed by JD Photo Data (UK), and exposed to UV light, baked and developed according to manufacturer instructions. SU8 grade and spincoat speeds are indicated in Supplementary Table 1. Poly-di-methylsiloxane (PDMS), in a 1:10 crosslinker to base ratio, was poured on the cast, degazed, baked

overnight at 70 °C, peeled off, cut to shape and covalently bound on isopropanol-rinsed microscopy glass slides after oxygen plasma treatment. PDMS to glass covalent bounds were allowed to form for 15 minutes at room temperature and the devices were then filled with sterile DI water for short term storage (few hours), in order to retain their hydrophilic properties.

**Chemotaxis assay.** The assay was described in detail previously[49]. In the micro-fluidic device (Fig. 1a), the two reservoirs were filled with cell suspensions of the same volume fraction supplemented with either no or 200 μM $\alpha$-methyl-D,L-aspartic acid (MeAsp), and then sealed to avoid residual flows. The MeAsp gradient formed in the 2 mm long, 1 mm wide, channel connecting them. Bacterial motion in the middle of the channel was recorded thrice, at one hour interval, at mid-height, in phase contrast microscopy at 10× magnification, using a Mikrotron 4CXP camera (1 px = 1.4 μm, field of view $A_0 = 512 \times 512$ px$^2$, 1 ms exposure) running at 100 frames/s (fps).

**Gradient calibration.** For gradient calibration experiments, the suspensions—at various cell densities—in the second reservoir were supplemented with 100 μM fluorescein as well as attractant. The gradient was measured in wide-field fluorescence microscopy (excitation filter 470/40, emission 525/50) at 10× magnification. Images were recorded using an Andor Zyla 4.2 sCMOS camera in the middle of the channel and in the attractant reservoir, the former being divided by the latter to correct for inhomogeneous illumination.

**Motility measurements by Fourier image analysis.** Cell motility was quantified by analysing the movies with three different algorithms. The first is differential dynamic microscopy (DDM), a now well established technique[46,47], computing differential image correlation functions (DICF), which were fitted to extract the average swimming speed of the population of cells $v_0$ and the fraction of swimming cells $\alpha$, as well as the local cell density (see below). The shortcomings due to the breakdown of two assumptions of Wilson et al.[46]—even distributions of swimming directions and round cell images—were accounted for via calibrations (see below). Second, phase differential microscopy (ΦDM)[49] determined the population-averaged drift velocity of the population of cells $v_d = \langle v_x(i,t) \rangle_{i,t}$, with positive $x$ being the up-gradient direction, $i$ cell index and $t$ frame number. The chemotactic velocity, corrected for the fraction of non swimming cells, is then $v_{ch} = v_d/\alpha$. Lastly, maps of the local velocity field were obtained by using a local ΦDM algorithm described in detail below. This image velocimetry technique estimated the velocity of a group of few cells located within a region 5 μm in diameter around a position $\mathbf{r} = (x, y)$. The local velocity field $\mathbf{v}(\mathbf{r}, t)$ was thus calculated for each time point $t$, and its spatial Fourier transform, $\tilde{\mathbf{v}}(\mathbf{q}, t) = \int\int d\mathbf{r}\, \mathbf{v}(\mathbf{r}, t) \exp(-i\mathbf{q}.\mathbf{r})$, then led to the flow structure factor via Eq. 1.

**Local image velocimetry algorithm.** The image analysis algorithm is as follows, with parameter values as chosen for our analysis: In a movie of $T = 10^4$ frames of size $A_0 = L_0 \times L_0$ pixels ($L_0 = 512$), the local velocities are computed at points situated on a square lattice spaced by a distance $da = 4$ px, starting from a distance $a/2$ from the edges of the frame ($a = 32$ px). For this, submovies of size $a \times a$ are considered, each centred on one of the points (represented by the index $k$ in the following). The spatial discrete Fourier transform of the blurred time frame $t$ in the submovie $k$ is computed as:

$$I_k^{bl}(\mathbf{q}, t) = \int\int dr I_k(\mathbf{r}, t) \exp(-\mathbf{r}^2/l^2) \exp(-i\mathbf{q}\mathbf{r}) , \qquad (5)$$

where $r = 0$ in the centre of the submovie and $l = 3$ px is a filtering range. The size of the group of cells, for which the instantaneous velocity is computed, can thus be tuned independently of the Fourier transform parameters. Changes in pixel intensities at a distance farther than $l$ will not contribute to the evaluation of the velocity at the point considered.

In the fashion of ΦDM[49], the phase $\delta\phi_k(\mathbf{q}, t)$ of the complex correlator $I_k^{bl}(\mathbf{q}, t+1)I_k^{bl}(\mathbf{q}, t)^*$ is computed and summed to get $\phi_k(\mathbf{q}, t) = \sum_{t'=0}^{t-1} \delta\phi_k(\mathbf{q}, t')$. This phase is fitted as a function of $\mathbf{q}$ to get the cummulated displacement $\mathbf{r}_k(t)$, $\phi_k(\mathbf{q}, t) = \mathbf{q} \cdot \mathbf{r}_k(t)$[49]. The local instantaneous velocity $\mathbf{v}_k(t)$ is then computed by fitting:

$$\mathbf{r}_k(t') = \mathbf{r}_k(t) + \mathbf{v}_k(t)(t' - t) \qquad (6)$$

on a range of $\tau = 20$ frames centred on $t$. Note that this method for measuring local displacements is equivalent to particle image velocimetry (PIV), since the phase of the correlation in Fourier space corresponds to the peak of the image cross-correlation in real space used in PIV to measure such displacement.

**In situ measurement and calibration of cell body volume fraction.** Chemotaxis microfluidic chips without a chemoattractant gradient were filled with suspensions of defined cell concentration, measured by OD$_{600}$. The DCIF $g(q, \tau) = \langle |I(\mathbf{q}, t+\tau) - I(\mathbf{q}, t)|^2 \rangle_{t,|\mathbf{q}|=q}$ were computed from DDM and fitted according to

$$g(q, \tau) = a_0(q) + a_1(q)(1 - f(q, \tau)) , \qquad (7)$$

where $f(q, \tau)$ is the intermediate scattering function yielding the cells average swimming speed and fraction of swimmers[46,47]. The amplitude $a_1 \simeq g(q, +\infty) = 2\langle |I(\mathbf{q}, t)|^2 \rangle_{t,|\mathbf{q}|=q}$ is expected to scale as $a_1(q) \sim f(N_{part})\langle I \rangle^2 F(q) S(q)$, where $N_{part}$ is the number of bacteria in the field of view, $S(q)$ is the structure factor of the bacterial fluid and $F(q)$ is a form factor describing the shape of the bacteria. As can be seen in Supplementary Fig. 2a, $a_1(q)/\langle I \rangle^2$ has a single maximum for each OD$_{600}$. This maximum was found to obey the equation $a_1^{max}/\langle I \rangle^2 = 1.25\, m\, OD_{600}/(1 + m\, OD_{600})$ (Supplementary Fig. 2b), where $m$ is a constant depending on the length of the bacteria and the height of the channel, for channel heights $h = 50$ μm and $h = 30$ μm. In the channel height $h = 8$ μm, the data were more scattered due to the difficulty of obtaining a truly homogeneous suspension even in absence of gradient, and the equation $a_1^{max}/\langle I \rangle^2 = d_0\, m\, OD_{600}/(1 + m\, OD_{600})$, with $d_0$ also depending on cell length, was found to better fit the data (Supplementary Fig. 2c). In all experiments, we then used $a_1^{max}/\langle I \rangle^2$ obtained from the DDM measurement to evaluate the local OD$_{600}$ via the appropriate equation.

The correspondance between optical density and cell body volume fraction was determined by counting cells in suspensions of known OD$_{600}$ flowing at 17.5 μL/min in a flow cytometer (BD Fortessa), which led to a size-dependent correspondance between cell number density ($n_c$) and OD$_{600}$. The average length $L$ of the cells was evaluated by segmenting and fitting as ellipses phase contrast microscopy images of the cells (magnification × 40, NA 0.95) using ImageJ (Rasband, W.S., ImageJ, U. S. National Institutes of Health, Bethesda, Maryland, USA, http://imagej.nih.gov/ij/, 1997–2016.). The cell body volume fraction was then $\Phi_c = \pi r^2 L n_c$, where $r = 0.5$ μm is the radius of an E. coli cell. The volume fraction was found to be proportional to optical density for both cell sizes, $\Phi_c = (1.78 \pm 0.03)\, 10^{-3}\, OD_{600}$ (Supplementary Fig. 2d), yielding the local $\Phi_c$ for all experiments.

**Calibrations of the swimming speed measurement**
*Confinement effect compensation.* Classical fits of the intermediate scattering function[46] assume that the cells are swimming in 3D with equal probability in all directions. Biased swimming directions in the plane of the field of view were also shown to not affect the measurement as long as the swimming speed of the cells is the same in all directions[49] (in other words, if chemotaxis, but not chemokinesis, biases swimming). If the distribution of swimming directions is however biased in the direction perpendicular to the field of view (z), the measured velocity will be systematically biased. This was notably the case when the suspension gets confined towards two dimensions, in the $h = 30$ and 8 μm channels. Analytical predictions exist for the DICF both for the 3D and 2D cases, but intermediate situations are more complex; The 3D analytical expression was thus used to extract the velocity and corrected as follows.

Since cells are partially confined in the z direction, we expect that the swimming speed $v_0$ measured using the formula, valid for cells swimming isotropically in three dimensions[46]:

$$f(q, \tau) = e^{-Dq^2\tau}\left(1 - \alpha + \alpha\left(\frac{Z+1}{Zqv_0\tau}\right)\frac{\sin(Z^{-1}\tan^{-1}\lambda)}{(1+\lambda^2)^{Z/2}}\right) , \qquad (8)$$

with $\sigma_v = v_0/\sqrt{Z+1}$ and $\lambda = qv_0\tau/(Z+1)$), will be systematically biased compared to the real velocity. To evaluate this bias, the following experiment was conducted. The PDMS chips with 50 and 30 μm height were bound by plasma treatment facing each other and to a glass slide (Supplementary Fig. 11a). Cell suspensions at OD$_{600} \simeq 1$ were flown in the chamber—without a gradient created—and let to equilibrate for one hour. The motion of the bacteria was recorded successively in the two channels of different heights and the effective swimming speed $v_0$ was extracted using Eqs. 7 and 8 for both channels. The ratio $v_0(30)/v_0(50)$ was found to be $1.075 \pm 0.01$ (SD − $N = 10$ repeats). The ratio being higher than 1 was expected since confinement makes the horizontal runs more probable and therefore increases the apparent 3D velocity. The same game played on the $h = 8$ μm channel lead to a ratio $v_0(8)/v_0(50) = 1.14 \pm 0.02$ (SD − $N = 5$ repeats). All effective velocities obtained by 3D fitting in both devices of lower heights were divided by the corresponding factor to deduce the real swimming speed.

*Effect of cell length.* The formula 7 is valid only in the case of bacteria with isotropic shapes or for which the orientation of the anisotropic shape is uncorrelated with the direction of motion. This is a good approximation in the case of 2 μm long cells observed in phase contrast at 10× magnification, but not anymore for 4 μm long cells in these conditions. For anisotropic objects like rods, Eq. 7 becomes, in the non interacting case:

$$g(q, \tau) = a_0(q) + \langle I^2 \rangle S(q) \int_0^{2\pi} \langle F(q, \theta)f(q, \theta, v\tau) \rangle d\theta \qquad (9)$$

were $\theta$ measures the orientation of the rods. Only when $q$ becomes smaller than $1/a$, $a$ being the largest size of the objects, does $F(q, \theta) \rightarrow 1$, and $g(q, \tau)$ can be fitted as in Wilson et al.[46] (Eqs. 8 and 7) irrespective of the shape of the particles. The outcome $v_0(q)$ of the fit of the DICF is plotted for randomly chosen experiments with non-treated and elongated cells in Supplementary Fig. 11b. For

elongated cells, above $q = 0.9$ px$^{-1}$, the fitted $v_0$ decreased because of the anisotropy. We therefore used the values of $v_0$ in the range $q = 0.4 - 0.9$ px$^{-1}$ to evaluate the swimming speed in the case of elongated cells, whereas $q = 0.4 - 2.0$ px$^{-1}$ was used for the normal cells.

**Simulations**. We performed agent-based simulations of the rod-like—length L, width e—chemotactic particles in two dimensions in a $256 \times 256$ μm$^2$ box with periodic boundary conditions. The particles are self-propelled in an overdamped fluid at constant propulsion force and interacted by exerting forces and torques on each other in the form of a Hertzian repulsion $\mathbf{F}_{ij}^{el}$, a friction $\mathbf{F}_{ij}^{fr}$ and far-field hydrodynamic interactions, which we accounted for following Jeckel et al.[8]. We write $\mathbf{F}_{ij}^{tot} = \mathbf{F}_{ij}^{el} + \mathbf{F}_{ij}^{fr}$ the sum of the elastic repulsion and friction generated upon contact by particle $j$ on $i$:

$$\mathbf{F}_{ij}^{el} = K_{el} \, \delta_{i,j}^{3/2} \, \mathbf{u}_{ij} \, , \tag{10}$$

$$\mathbf{F}_{ij}^{fr} = -K_{fr}(\mathbf{v}_i - \mathbf{v}_j) \cdot \mathbf{v}_{ij} \, \mathbf{v}_{ij} \, , \tag{11}$$

with $\mathbf{u}_{ij}$ and $\mathbf{v}_{ij}$ the normalized orthogonal vectors defined in the inset of Fig. 4a, $\delta_{i,j}$ the inter-penetration depth between particle $i$ and $j$. The free running velocity of particle $i$ is $\mathbf{v}_i = v_i \mathbf{n}_i$, with a norm $v_i$ chosen randomly in a Gaussian distribution of mean $v_0$ and variance $dv_0$ and direction $\mathbf{n}_i = (\cos(\theta_i), \sin(\theta_i))$. Note that $\mathbf{F}^{fr}$, which was not included in previous studies[22–24], could be interpreted as effectively resulting from short-range cell-cell interactions including viscous shear in the thin liquid film between two cells, complex flagellar entanglement, and solid friction. The particles can be in two states, either running ($\Theta_i = 1$) or tumbling ($\Theta_i = 0$). During tumbles, self-propulsion is switched off and a random torque made the particle turn, with an average magnitude chosen to match literature reported turning angle distributions[74]. The equations of motion for their position $\mathbf{r}_i$ and orientation $\theta_i$ are therefore:

$$\frac{d\mathbf{r}_i}{dt} = \Theta_i \, v_i \mathbf{n}_i + \Sigma_j \frac{\mathbf{w}_j(\mathbf{r}_i) + \mathbf{w}_j(\mathbf{r}_i - L_{dip}\mathbf{n}_i)}{2} + (\gamma_\perp^{-1}\mathbf{I} + (\gamma_\parallel^{-1} - \gamma_\perp^{-1})\mathbf{n}_i\mathbf{n}_i)\Sigma_j\mathbf{F}_{ij}^{tot} \tag{12}$$

$$\frac{d\theta_i}{dt} = \sqrt{D_r}\eta_r(t) + \sqrt{D_T}\eta_T(t) \, (1 - \Theta_i)$$
$$+ \mathbf{n}_i \wedge \Sigma_j \left( \frac{\mathbf{w}_j(\mathbf{r}_i) - \mathbf{w}_j(\mathbf{r}_i - L_{dip}\mathbf{n}_i)}{L_{dip}} + \gamma_r^{-1}l_{i,j}\mathbf{F}_{ij}^{tot} \right) . \tag{13}$$

The translational friction coefficients read $\gamma_\parallel = 2\pi\eta L/\ln(L/e)$ and $\gamma_\perp = 2\gamma_\parallel$, and $\gamma_r = \gamma_\parallel L^2/6$ is the rotational friction coefficient. The distance between the centre of mass of the particle and the contact point with particle $j$ is $l_{i,j}$. Rotational Brownian motion is modelled using the rotational diffusion coefficient $D_r$ and the normal Gaussian noise $\eta_r(t)$, and reorientation during tumbles using corresponding $D_T$ and $\eta_T(t)$.

When hydrodynamic interactions are included, $\mathbf{w}_j(\mathbf{r})$ is the flow generated by particle $j$ at point $\mathbf{r}$. The particles thus react to external flows as a dumbell. As a source of flow, they are modelled as a dipole of point forces of pusher symmetry (Fig. 4a). The point of application of the resistive force coincides with the centre of the rod, the one of the propulsive force being situated $L_{dip} = L/2 + L_{flag}/2$ behind the latter in the direction of the rod. Following Jeckel et al.[8], we used a quasi-2D approximation to compute the flow generated by the 2D point force $\mathbf{F} = \gamma_\parallel v_j \mathbf{n}_j$. The fluid velocity $\mathbf{u}$ is assumed to take a Poiseuille profile in the z-direction, its average (x,y)-component $\mathbf{U} = 1/h \int_0^h \mathbf{u}(x,y,z)dz$ thus obeying an effective 2D Hele-Shaw equation:

$$\nabla \cdot \mathbf{U} = 0 \tag{14}$$

$$\nabla P - \eta(\nabla^2\mathbf{U} - \kappa\mathbf{U}) = \frac{\mathbf{F}}{h}\delta^{(2)}(\mathbf{r}) \tag{15}$$

where $P$ is the pressure, $\nabla = (\partial_x, \partial_y)$ and $\kappa = 12/h^2$, the origin being taken at the point of application of $\mathbf{F}$. This equation was solved in Jeckel et al.[8]:

$$\mathbf{U} = \mathbf{G} \cdot \mathbf{F} \tag{16}$$

with the symmetric matrix:

$$\mathbf{G} = \frac{1}{\pi\eta h \, \kappa|r|^2} \left[ c_1(\sqrt{\kappa}|r|)\mathbf{I} + c_2(\sqrt{\kappa}|r|)\hat{\mathbf{r}}\hat{\mathbf{r}} \right] \tag{17}$$

where $\hat{\mathbf{r}} = \mathbf{r}/|r|$ and the factors:

$$c_1(z) = z^2[K_0(z) + K_2(z)]/2 - 1 \tag{18}$$

$$c_2(z) = 2 - z^2K_2(z) \tag{19}$$

where $K_n$ are modified Bessel functions of the second kind. The flow generated by particle $j$ at a position $\mathbf{r}$ from the centre of the rod is then:

$$\mathbf{w}_j(\mathbf{r}) = \mathbf{U}(\mathbf{F}_j, \mathbf{r}) + \mathbf{U}(-\mathbf{F}_j, \mathbf{r} + L_{dip}\mathbf{n}_j) \tag{20}$$

which we compute as such in the algorithm, because the dipole approximation do not model short-range flows accurately enough for our simulations. Note that the range of the interaction is set by the height of the channel via $\kappa$. Interestingly, the term $\eta\kappa\mathbf{U}$ which introduces this scaling in Eq. 15 comes from a term of friction on the wall $(-\eta(\partial_z\mathbf{u}(h) - \partial_z\mathbf{u}(0)))$ which appears when averaging the 3D Stockes equation over the channel height. The fluid flow is plotted for the three channel heights we used in Supplementary Fig. 6a. When the height increases, the flow field goes from circular flow patterns to a pattern qualitatively more akin to 3D dipoles (although quantitatively different).

Finally, the state of each particle ($\Theta_i$) was determined by an internal chemotaxis system evolving according to the ligand concentration experienced by the cells, first formulated in Vladimirov et al.[59], with the only difference that the instantaneous tumble probability was set directly using the concentration of phosphorylated CheY (CheY-P), disregarding the existence of multiple flagellar motors. This model is detailed in Supplementary Note 2.

The set of equations 10–13, 16–20 and Supplementary Eqs. 1–8, which governs the position and orientation of the rods, was solved by Euler integration. Parameter values[59,75] are given in Supplementary Table S2.

## Data availability
The data supporting the findings of this study, including raw images and movies, are available upon request from the corresponding authors.

## Code availability
The source codes of the image analysis methods are available at https://github.com/croelmiyn/FourierImageAnalysis under MIT Licence and can be accessed and cited using https://doi.org/10.5281/zenodo.3516258. The code of the simulations is available at https://github.com/croelmiyn/SimulationsChemotaxis under MIT Licence. The source code can be accessed and cited using https://doi.org/10.5281/zenodo.3516624. All codes were writen in Java v1.8.0 and implemented in the form of Plugins for ImageJ v1.49.

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

## Acknowledgements

The authors thank Prof. W. Stolz, M. Koch, and K. Volz for access to the micro-fabrication facility at the Center for Material Science (Philipps-Universität Marburg), Dr. L. Laganenka for providing the Δflu mutant and Hannah Jeckel for assistance with the hydrodynamic simulations. This work was supported by grant 294761-MicRobE from the European Research Council.

## Author contributions

R.C. and V.S. designed the research, R.C. performed and analyzed the experiments and simulations, K.D. contributed to simulations, R.C., K.D. and V.S. wrote the manuscript.

## Competing interests

The authors declare no competing interests.
