## [Peer Review File · Nature Communications]

REVIEWERS' COMMENTS:

Reviewer #5 (Remarks to the Author):

The manuscript by Colin et al. entitled "Chemotactic behaviour of Escherichia coli at high cell density" has very positive aspects to it, as well as a couple of shortcomings. Overall, I think it deserves publication in Nature Communications. This actually is mainly due to its novelty, as it addresses for the first time how chemotactic ability of bacteria is affected by cell density with high relevance to biology. Furthermore, the technical concerns raised by the other reviewers have been addressed adequately. In particular, both steric and hydrodynamic effects are investigated in the latest version of the manuscript.

Pros:

The overall mechanistic picture which emerges is that a peak of chemotaxis for intermediate cell densities is due to steric effects, while the decline at large densities is due to hydrodynamics (vortices dominating over chemotaxis).

Cons:

Would have preferred that the steric and hydrodynamic effects are investigated first separately, and then merged, maybe with some heuristic corrections to explain the data. Instead, the effects are all mixed together (but additional plots in SI are helpful) and due to limitations in the theory, it remains a bit unclear what works and what not. The theory limitations are:

- Simulations are in 2D only (but Hele-Shaw approximation is well established)
- Hydrodynamic theory works for far field only (dipole approximation of point forces of pusher symmetry)
- Drift formula corrected for cell-cell interactions by enhanced "rotational diffusion" but due to potential many-body effects this effective single-cell behaviour theory may well not be correct.
- Flagella interactions are not discussed. They could have been included heuristically by an additionally hypothesized cell-cell interaction term.
- Little details on how algorithm is computationally implemented, and no reference to code on Github or any other depository.

As some of these Cons are technically very difficult to remedy, the Pros clearly stand out, and the paper represents an important contribution to the field of collective chemotaxis.

REVIEWERS' COMMENTS:

Reviewer #5 (Remarks to the Author):

The manuscript by Colin et al. entitled "Chemotactic behaviour of Escherichia coli at high cell density" has very positive aspects to it, as well as a couple of shortcomings. Overall, I think it deserves publication in Nature Communications. This actually is mainly due to its novelty, as it addresses for the first time how chemotactic ability of bacteria is affected by cell density with high relevance to biology. Furthermore, the technical concerns raised by the other reviewers have been addressed adequately. In particular, both steric and hydrodynamic effects are investigated in the latest version of the manuscript.

Pros:

The overall mechanistic picture which emerges is that a peak of chemotaxis for intermediate cell densities is due to steric effects, while the decline at large densities is due to hydrodynamics (vortices dominating over chemotaxis).

Cons:

Would have preferred that the steric and hydrodynamic effects are investigated first separately, and then merged, maybe with some heuristic corrections to explain the data. Instead, the effects are all mixed together (but additional plots in SI are helpful) and due to limitations in the theory, it remains a bit unclear what works and what not. The theory limitations are:

- 1) Simulations are in 2D only (but Hele-Shaw approximation is well established)*
- 2) Hydrodynamic theory works for far field only (dipole approximation of point forces of pusher symmetry)*
- 3) Drift formula corrected for cell-cell interactions by enhanced "rotational diffusion" but due to potential many-body effects this effective single-cell behaviour theory may well not be correct.*
- 4) Flagella interactions are not discussed. They could have been included heuristically by an additionally hypothesized cell-cell interaction term.*
- 5) Little details on how algorithm is computationally implemented, and no reference to code on Github or any other depository.*

As some of these Cons are technically very difficult to remedy, the Pros clearly stand out, and the paper represents an important contribution to the field of collective chemotaxis.

We thank Reviewer #5 to take the time to assess our work and to find that it is suitable for publication in *Nature communications*. We would like to briefly address the points he/she raised below.

1-2) In our simulations, steric interactions were always included, but hydrodynamics could either be on or off. In the hydrodynamics-only case, forces diverge when two cells come unphysically close from each

other. Preventing cells to come this close is by definition introducing a steric interaction. This is why a hydrodynamics-only model was not studied. We emphasized strongly notably in the discussion that hydrodynamic interactions are necessary to reproduce the characteristics of the collective motion, but that the decrease of the chemotactic drift comes from forced reorientations, whether they are caused by steric or hydrodynamic interactions. In our experimental system, these reorientations come from the latter, but in other systems like swarms, steric effects could be more prominent.

Although our simulations included all the relevant physics of the system (hydrodynamics, steric collisions, chemotactic pathway response), some level of simplification was unavoidable. As the Reviewer pointed out, going beyond the force dipole and 2D Hele-Shaw approximations for the hydrodynamics part is technically very challenging, and would constitute a study on its own, especially when individual chemotactic behavior is considered at the same time. As mentioned before, steric interactions, by keeping cells apart, prevent the dipole approximation, which is indeed far-field, from becoming too unphysical.

3) As the reviewer correctly assessed, we actually show in Supplementary Figures 9d and 10d that treating the effect of cell-cell interactions as an effective enhancement of rotational diffusion becomes indeed invalid at high cell density, although the deviations between the simulation and the “effective” theory are of 40% at their maximum. This is mentioned at the end of the result section.

4) Since we do not have a reasonable model for the effect of flagellar entanglement, we decided to not include it specifically in the simulations. We nevertheless added a friction term under contact between two bacteria, which could be interpreted as an effective account of several short range interactions including flagellar entanglement, as we discuss in the relevant Methods section.

5) We now mention that all software was written in Java 1.8.0 and implemented in the form of Plugins for ImageJ v1.49. Codes have now been made available on GitHub, and can be accessed and cited using, for the image analysis software, DOI:10.5281/zenodo.3515924 and, for the model, DOI: 10.5281/zenodo.3516624